# Three New Species of *Microdochium* (*Microdochiaceae*, *Xylariales*) on *Bambusaceae* sp. and Saprophytic Leaves from Hainan and Yunnan, China

**DOI:** 10.3390/jof9121176

**Published:** 2023-12-07

**Authors:** Jie Zhang, Zhaoxue Zhang, Duhua Li, Jiwen Xia, Zhuang Li

**Affiliations:** Shandong Provincial Key Laboratory for Biology of Vegetable Diseases and Insect Pests, College of Plant Protection, Shandong Agricultural University, Taian 271018, China; zhjie8087@163.com (J.Z.); zhangzhaoxue2022@126.com (Z.Z.); ldh3458198584@163.com (D.L.); xiajiwen1@126.com (J.X.)

**Keywords:** *Microdochium*, multigene phylogeny, new taxon, taxonomy

## Abstract

Species of the genus *Microdochium* (*Microdochiaceae*, *Xylariales*) have been reported from the whole world and separated from multiple plant hosts. The primary aim of the present study is to describe and illustrate three new species isolated from the leaf spot of *Bambusaceae* sp. and saprophytic leaves in Hainan and Yunnan provinces, China. The proposed three species, viz., *Microdochium bambusae*, *M. nannuoshanense* and *M. phyllosaprophyticum,* are based on multi-locus phylogenies from a combined dataset of ITS rDNA, LSU, RPB2 and TUB2 in conjunction with morphological characteristics. Descriptions and illustrations of three new species in the genus are provided.

## 1. Introduction

*Microdochium* Syd. & P. Syd., belonging to *Microdochiaceae* Hern.-Restr., Crous & J.Z. Groenew., was introduced by Syd. & P. Syd. in 1924 [1]. *Microdochium phragmitis* Syd. & P. Syd., the genus type, was introduced by Sydow on leaves of *Phragmites australis* in Germany in 1919 [1]. *Microdochium* species had frequently been separated into endophytes, plant pathogens and saprophytes, and were commonly isolated from some diseased plant hosts [2,3,4,5,6,7]. At present, about 62 species of *Microdochium* are listed in the Index Fungorum (http://www.indexfungorum.org/, accessed on 30 October 2023). Species of *Microdochium* are characterized by coelomycetous asexual morphs producing polyblastic, sympodial or annellidic conidiogenous cells with hyaline conidia. Sexual morphs are *Monographella*-like with present or absent stromata, perithecial ascomata, eight-spored, oblong, clavate asci, apical ring, and hyaline to pale brown, fusoid ellipsoid or oblong ascospores [8].

Previous studies have shown that *Microdochium* belongs to *Amphisphaeriaceae,* based on its morphological resemblance [9,10,11,12]. Hernández-Restrepo et al. [8] suggested that *Idriella* and *Microdochium* could be congeneric. In their phylogenetic analysis, *Idriella*, *Microdochium* and *Selenodriella* clustered in *Xylariales* as a distinct monophyletic lineage. Therefore, Hernández-Restrepo et al. [8] introduced the new family *Microdochiaceae* to accommodate this clade. The hosts of *Microdochium* are diverse and widely distributed [8,13,14,15,16,17,18]. In recent years, *Microdochium* has included important plant pathogens; for example, Kwasna et al. [19] gave an up-to-date description of *Microdochium triticicola,* which was split from the roots of *Triticum aestivum* L. in the UK. Zhang et al. [20] identified *Microdochium paspali*, which caused leaf blight of *Paspalum vaginatum* Sw., a widely used lawn grass for tropical and subtropical golf courses. Liu et al. [21] described three species, *Microdochium miscanthi*, *M. sinense* and *M. hainanense,* isolated from *Miscanthus sinensis* Anderss. and *Phragmites australis* (Cav.) Trin. ex Steud in Hainan, China. Dissanayake et al. [22] described *M. sichuanense* isolated from a *Poaceae* host in Sichuan, China. Liang et al. [2] described new species *Microdochium poae*, which caused leaf blight of Creeping bentgrasses (*Agrostis stolonifera* L.) and Kentucky blue grass (*Poa pratensis* L. var. anceps Gaud.). In particular, Mandyam et al. [5] sporulated dark septate endophytes from roots of mixed tallgrass prairie plant communities, and sporulated species of *Aspergillus*, *Fusarium*, *Microdochium* and *Periconia* by ITS-RFLP and/or sequencing of the internal transcribed spacer of the ribosomal RNA gene (ITS) region. In addition, *Microdochium* produced abundant melanized inter- and intracellular chlamydospores.

Fungi associated with leaf spots were collected from *Bambusaceae* sp. and saprophytic leaves. Morphological characteristics were obtained by separation and purification. The sequences of four molecular markers, viz., the partial nuclear ribosomal large subunit (LSU), the ITS gene, the partial RNA polymerase II second-largest subunit (RPB2) and the partial β-tubulin gene (TUB2), were used in this study. We identify these fungi as three species of the genus *Microdochium*, proposed herein.

## 2. Materials and Methods

### 2.1. Sample Collection, Fungal Isolation and Morphology

*Bambusaceae* sp. and saprophytic specimens showing necrotic spots were collected during a series of field visits in Hainan and Yunnan provinces in China in 2022–2023. A specimen usually isolates multiple fungi, and we obtained a single colony through single spore isolation and tissue isolation methods [23]. We removed fragments (4 × 4 mm) from the damaged edges of the leaves and disinfected the surface by continuously soaking them in 75% ethanol solution and rinsing them in sterile distilled water and 10% sodium hypochlorite solution, and then we rinsed them three times in sterile deionized water for 60 s, 45 s, 45 s and 30 s, respectively. We placed the processed fragments on sterile filter paper to absorb moisture, and then placed them on Potato Dextrose Agar (PDA potato: 200 g, dextrose: 15 g, agar: 15 g, distilled water 1 L and natural pH) or Oat Meal Agar (OA: oats: 30 g, agar: 15 g, distilled water: 1 L and natural pH) and incubated them at 24 °C for 3–5 days. Then, the agar portion containing fungal hyphae from the periphery of the colony was transferred to a new PDA plate and was photographed using a Sony ZV-E10L digital camera (Sony Group Corporation, Tokyo, Japan) on days 5, 10 and 15. Using an Olympus SZ61 stereo microscope and an Olympus BX43 microscope (Olympus Corporation, Tokyo, Japan), respectively, in conjunction with an Olympus DP73 and OPTIKA SC2000 high-definition color digital camera, the microscopic morphological characteristics of the structures produced in the culture were observed to capture and record fungal structures. All fungal strains were stored in 15% sterilized glycerol at 4 °C, with each strain stored in three tubes (2.0 mL tubes), for further research. Digimizer software (v5.6.0) was used for structural measurements, with 25 or more measurements taken for each character (conidiophores, conidiogenous cells, conidia and so on) [23]. Specimens were deposited in the HSAUP and HMAS [23]. Living cultures were deposited in the SAUCC [23].

### 2.2. DNA Extraction, PCR and Sequencing

Fungal DNA was extracted from the fresh mycelia grown on PDA or OA using a CTAB (cetyltrimethylammonium bromide) method and a kit method (OGPLF-400, GeneOnBio Corporation, Changchun, China) [24,25]. Four molecular markers, including LSU, ITS, RPB2 and TUB2 gene, were amplified with the primer pairs listed in Table 1. An amplification reaction was carried out at 20 μL reaction volume, including 10 μL 2 × Hieff Canace^®^ Plus PCR Master Mix (Shanghai, China) (with dye) (Yeasen Biotechnology, Shanghai, China, Cat No. 10154ES03), 0.5 μL each of forward and reverse primer, and 0.5 μL template genomic DNA, adjusted to a total volume of 20 μL using distilled deionized water. Some 1% agarose gel and GelRed (TsingKe, Qingdao, China) were used to separate and purify the PCR product, and ultraviolet light was used to observed whether the fragment was consistent. Then a Gel Extraction Kit (Cat: AE0101-C) (Shandong Sparkjade Biotechnology Co., Ltd., Jinan, China) was used for gel recovery. The PCR products were processed for purification and bidirectional sequencing by TsingKe Biological Technology, Qingdao, China. The raw data (trace data) were processed using MEGA v. 7.0 to obtained consistent sequences, including removing disordered peak sequences and complementary concatenation of forward and reverse sequences (ClustalW) [26]. All sequences generated in this study were deposited in GenBank under the accession numbers in Table 2.

### 2.3. Phylogeny

The newly generated sequences in this study and closely associated sequences from Liu et al. [21] and Dissanayake et al. [22] were aligned using the MAFFT 7 online service with the default strategy and corrected manually using MEGA 7 [26,31]. To determine the identity of the isolates at species level, phylogenetic analysis was first conducted separately for each marker, followed by a combination (ITS-LSU-RPB2-TUB2) (See Appendix A).

Phylogenetic analysis of multi-labeled data was based on Bayesian inference (BI) and maximum likelihood (ML) algorithms. Firstly, MrModeltest v. 2.3 [32] was used determine the best evolutionary model for each partition under the Akaike Information Criterion (AIC), which was used to identified the best nucleotide substitution model settings prior to the BI analysis. Secondly, ML and BI were run on the CIPRES Science Gateway portal (https://www.phylo.org/, accessed on 30 October 2023) or offline software (ML was operated in RaxML-HPC2 on XSEDE v8.2.12, and BI analysis was operated in MrBayes v3.2.7a with 64 threads on Linux) [33,34,35,36,37]. Thirdly, for ML analyses, the default parameters were used and 1000 rapid bootstrap replicates were run with the GTR+G+I model of nucleotide evolution; BI analysis was performed using a fast bootstrap algorithm with an automatic stop option. Finally, all resulted trees were plotted using FigTree v. 1.4.4 (http://tree.bio.ed.ac.uk/software/figtree, accessed on 20 October 2023) or ITOL: Interactive Tree of Life (https://itol.embl.de/, accessed on 20 October 2023) [38], and the layout of the trees was produced in Adobe Illustrator CC 2019.

## 3. Results

### 3.1. Phylogenetic Analyses

The comparison contained 58 isolates representing *Microdochium* and related taxa, and the used strains CBS 204.56 and CBS 177.57 of *Idriella lunata* were used as an outgroup. The final alignments consisting of 2945 characters were used for phylogenetic analyses, viz., 1–581 (ITS), 582–1418 (LSU), 1419–2258 (RPB2) and 2259–2945 (TUB2), including gaps. Of these characters, 2213 were constant, 68 were variable and parsimony-uninformative, and 664 were parsimony-informative. The topology of the ML tree was consistent with that of the Bayesian tree and was therefore considered to be representative of the evolutionary history of the genus *Microdochium* (Figure 1). The final ML optimization likelihood was −16,911.019760. The matrix had 681 distinct alignment patterns with 15.92% undetermined characters or gaps. Estimated base frequencies were as follows: A = 0.259370, C = 0.234527, G = 0.263997 and T = 0.242106; substitution rates were AC = 0.945501, AG = 4.719533, AT = 1.249092, CG = 0.946009, CT = 7.264699 and GT = 1.000000; and the gamma distribution shape parameter α = 0.124118. The Dirichlet base frequencies and the GTR+I+G evolutionary mode were used for ITS, K80+I+G was used for LSU, HKY+I+G was used for RPB2, and GTR+I was used for TUB2. MCMC analysis of these four tandem genes was performed over 245,000 generations in 4902 trees. The first 1224 trees representing the aging phase of the analysis were discarded, while the remaining trees were used to calculate the posterior probability in the majority-rule consensus tree (Figure 1; first value: PP > 0.80 shown). The alignment embodied a total of 848 unique site patterns (ITS: 237, LSU: 94, RPB2: 350 and TUB2: 167).

The 58 strains were assigned to 37 species clades on the phylogram (Figure 1). Among them, six strains conducted three new species lineages: *Microdochium bambusae* (SAUCC 1862-1, SAUCC 1866-1) was closely related to *M. indocalami* (SAUCC 1016) with good support (1.0 BIPP and 90% MLBV), *M. nannuoshanense* (SAUCC 2450-1 and SAUCC 2450-3) was closely related to *M. sinense* (SAUCC 211097 and SAUCC 211098) with full support (1.0 BIPP and 99% MLBV), and *M. phyllosaprophyticum* (SAUCC 3583-1 and SAUCC 3583-6) formed a separate single-species lineage. Based on the phylogenetic resolution and morphological analyses, three new species of the *Microdochium* species, viz., *Microdochium bambusae* sp. Nov., *M. nannuoshanense* sp. Nov. and *M. phyllosaprophyticum* sp. nov., were reported in the present study.

**Figure 1 jof-09-01176-f001:**
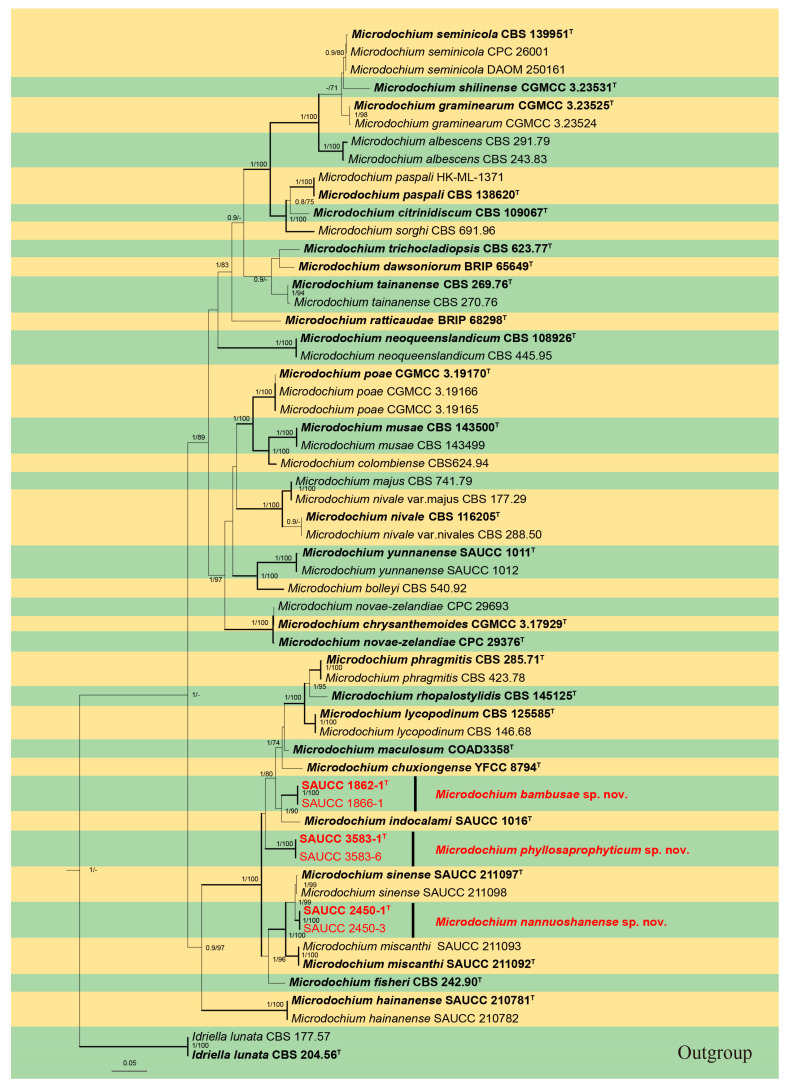
A maximum likelihood tree based on combined dataset of analyzed ITS, LSU, RPB2 and TUB2 sequence. Left, BIPP ≥ 0.80 and right, MLBV ≥ 70% are shown as BIPP/ML above the nodes. The branches BIPP/ML with 1/100 are indicated in bold. Ex-type cultures are indicated in bold face and marked with “^T^”. Strains from the present study are in red. The tree was rooted in *Idriela lunata* (CBS 204.56* and CBS 177.57). The yellow and green areas are used to distinguish different species. The scale bar at the bottom middle indicates 0.05 substitutions per site.

### 3.2. Taxonomy

#### 3.2.1. *Microdochium bambusae* J. Zhang, Z.X. Zhang, & Z. Li, sp. Nov.; Figure 2

MycoBank—No. MB850598

Etymology—Referring to the generic name of the host plant *Bambusaceae* sp.

Type—China, Yunnan Province: Xishuangbanna Tropical Botanical Garden, Chinese Academy of Sciences; on diseased leaves of *Bambusaceae* sp.; 15 March 2023; Z. X. Zhang (HMAS 352651, holotype); ex-holotype living culture SAUCC 1862-1.

Description—Endogenic on diseased leaves of *Bambusaceae* sp. Sexual morphs are unknown. Mycelia are superficial and immersed, 2.3–3.6 µm wide, branched, membranous and hyaline. Conidiophores are inapparent and often reduced to conidiogenous cells. Conidiogenous cells are straight or slightly curved, 17.4–30.0 × 2.5–3.0 µm, mono- or polyblastic, terminal, hyaline, septate, cylindrical and smooth, and produced on aerial mycelia. Conidia are solitary, hyaline, oblong to ellipsoid, straight or curved, 13.0–17.0 × 2.5–3.5 µm, multi-guttulate and sometimes borne directly from hyphae. Chlamydospores were not observed; see Figure 2.

Culture characteristics—Cultures incubated on PDA at 25 °C in darkness, reaching 55–60 mm diam., had a growth rate of 3.9–4.3 mm/day after 14 days, with moderate aerial mycelia, were gray white with regular margins, and the reverses were dark brown in the center and light brown to white at the edge. Cultures incubated on PDA at 25 °C in darkness, reaching 82–87 mm diam., had a growth rate of 5.8–6.2 mm/day after 14 days, with moderate aerial mycelia on the surface, were gray white with regular margins, and the reverses were similar in color.

Additional specimen examined—China, Yunnan Province: Xishuangbanna Tropical Botanical Garden, Chinese Academy of Sciences; on diseased leaves of *Bambusaceae* sp.; 15 March 2023; J. Zhang (HSAUP 1866-1); living culture SAUCC 1866-1.

Notes—Phylogenetic analyses of four combined genes (ITS, LSU, RPB2 and TUB2) showed that *Microdochium bambusae* sp. nov. formed an independent clade closely related to *M. indocalami. M. bambusae* is distinguished from *M. indocalami* (SAUCC 1016) by 7/519, 3/836, 59/840 and 19/715 characters in ITS, LSU, RPB2 and TUB2 sequences, respectively. Morphologically, *M. bambusae* differs from *M. indocalami* in conidia (13.0–17.0 × 2.5–3.5 μm vs. 13.0–15.5 × 3.5–5.5 μm) and in colony texture (gray white with regular margins vs. white with regular margins on PDA), and *M. indocalami* has more aerial mycelia than *M. bambusae* [17]. For details, see Table 3.

**Figure 2 jof-09-01176-f002:**
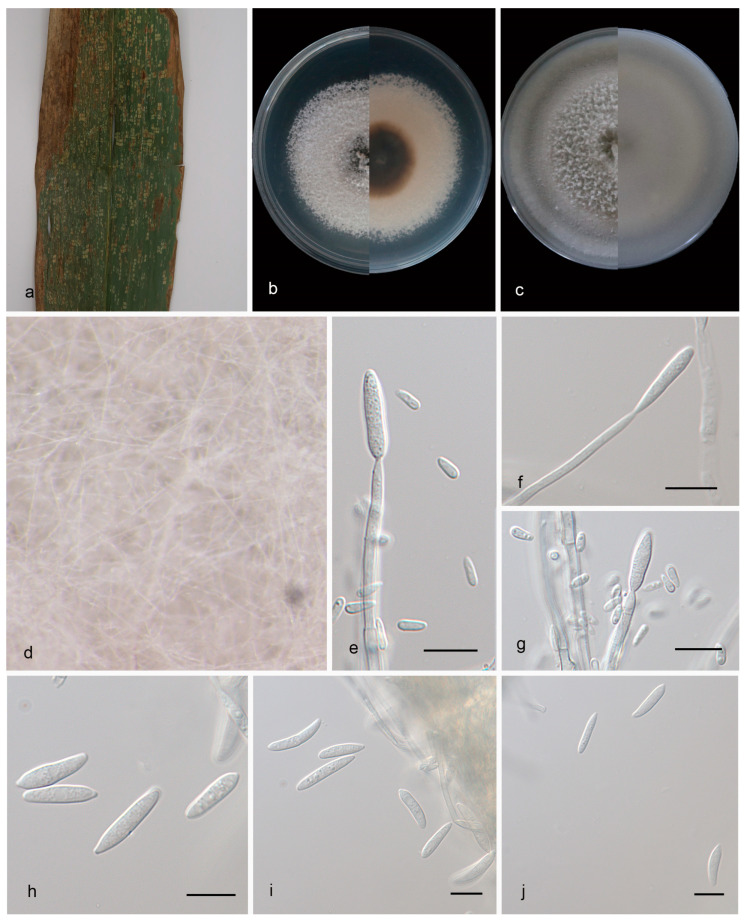
*Microdochium bambusae* (HMAS 352651, holotype). (**a**) A leaf of *Bambusaceae* sp.; (**b**) colonies on PDA from above and below after 14 days; (**c**) colonies on OA from above and below after 14 days; (**d**) colony overview; (**e**–**g**) conidiogenous cells and conidia; (**h**–**j**) conidia. Scale bars: (**e**–**j**) 10 μm.

#### 3.2.2. *Microdochium nannuoshanense* J. Zhang, Z.X. Zhang, & Z. Li, sp. Nov.; Figure 3

MycoBank—No. MB850597

Etymology—Referring to the location of the holotype, Nannuoshan, Yunnan Province, China.

Type—China, Yunnan Province, Nannuoshan; on diseased leaves of *Bambusaceae* sp.; 12 April 2023; J. Zhang (HMAS 352652, holotype); ex-holotype living culture SAUCC 2450-1.

Description—Endogenic on diseased leaves of *Bambusaceae* sp. Sexual morphs are unknown. Mycelia are superficial and immersed, 1.8–2.6 µm wide, branched, membranous and hyaline. Conidiophores are inapparent and often reduced to conidiogenous cells. Conidiogenous cells are straight or slightly curved, 19.0–27.0 × 2.0–3.0 µm, mono- or polyblastic, terminal, denticulate, transparent, smooth, cylindrical and septate, and produced on aerial mycelia. Conidia are solitary, hyaline, spindle to rod-shaped, straight or curved, oblong to ellipsoid, 7.0–12.0 × 2.5–4.0 µm, multi-guttulate and sometimes borne directly from the hyphae. Chlamydospores were not observed; see Figure 3.

Culture characteristics—Cultures incubated on PDA at 25 °C in darkness, reaching 68–73 mm diam., had a growth rate of 4.8–5.2 mm/day after 14 days, were creamy white to pale brown with regular margins, had moderate aerial mycelia, and the reverse was similar. Cultures incubated on OA at 25 °C in darkness, reaching 53–57 mm diam., with a growth rate of 3.7–4.1 mm/day, were creamy white with regular margins, had luxuriant aerial hyphae, and the reverse was similar.

Additional specimen examined—China, Yunnan, Nannuoshan; on diseased leaves of *Bambusaceae* sp.; 12 April 2023; J. Zhang; HSAUP2450-3; living culture SAUCC 2450-3.

Notes—Phylogenetic analyses of four combined genes (ITS, LSU, RPB2 and TUB2) showed that *Microdochium nannuoshanense* sp. nov. formed an independent clade closely related to *M. sinense*. *M. nannuoshanense* is distinguished from *M. sinense* (SAUCC 211097) by 4/535, 3/836, 16/909 and 4/717 characters in ITS, LSU, RPB2 and TUB2 sequences, respectively. Morphologically, *M. nannuoshanense* differs from *M. sinense* in conidia (7.0–12.0 × 2.5–4.0 μm vs. 11.5–19.34 × 2.8–5.4 μm) and in colony texture (creamy white to pale brown with regular margins vs. yellow-brown overall and fluffy at the edge on PDA) [21]. For details, see Table 3.

**Figure 3 jof-09-01176-f003:**
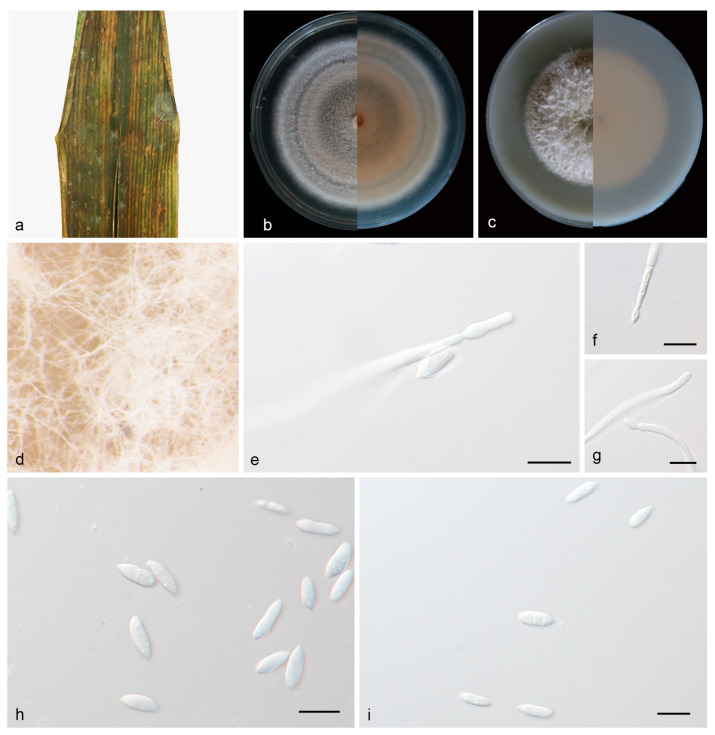
*Microdochium nannuoshanense* (holotype, HMAS 352652). (**a**) A leaf of *Bambusaceae* sp; (**b**) colonies on PDA from above and below after 14 days; (**c**) colonies on OA from above and below after 14 days; (**d**) colony overview; (**e**–**g**) conidiogenous cells and conidia; (**h**,**i**) conidia. Scale bars: (**e**–**i**) 10 μm.

#### 3.2.3. *Microdochium phyllosaprophyticum* J. Zhang, Z.X. Zhang, & Z. Li, sp. Nov.; Figure 4

MycoBank No. MB850599

Etymology—Referring to the generic name of the host plant’s saprophytic leaves (Greek “phyllum” and Latin “*sapro*[phyticum]”).

Type—China, Hainan Province: Bawangling National Forest Park; on saprophytic leaves; 11 April 2023; J. Zhang; holotype HMAS 352653; ex-holotype living culture SAUCC SAUCC 3583-1.

Description—Saprophytic on dead leaves. Sexual morphs are unknown. Mycelia are superficial and immersed, 2.4–3.3 µm wide, branched, membranous and hyaline. Conidiophores are inapparent and often reduced to conidiogenous cells. Conidiogenous cells are straight or slightly curved, 16.0–32.0 × 2.0–2.5 µm, mono- or polyblastic, terminal, denticulate, hyaline, smooth, cylindrical and membranous, and produced on aerial mycelia. Conidia are solitary, hyaline, cylindrical, oblong to ellipsoid, spindle to rod-shaped, straight or curved, 7.5–15.0 × 2.0–3.5 µm, multi-guttulate and sometimes borne directly from hyphae. Chlamydospores were not observed; see Figure 4.

Culture characteristics—Cultures incubated on PDA at 25 °C in darkness, reaching 60.0–63.0mm diam., had a growth rate of 4.3–4.5 mm diam/day after 14 days, were creamy white with regular margins, had luxuriant aerial hyphae, and the reverses were light brown in the center and white at the edge. Cultures incubated on PDA at 25 °C in darkness, reaching 62.0–68.0 mm diam., had a growth rate of 4.4–4.8 mm diam/day after 14 days, were creamy white with regular margins, had luxuriant aerial hyphae, and the reverses were similar.

Additional specimen examined—China, Hainan Province: Bawangling National Forest Park; on saprophytic leaves; 11 April 2023; J. Zhang; HSUAP3583-6; living culture SAUCC SAUCC 3583-6.

Notes—Phylogenetic analyses of four combined genes (ITS, LSU and RPB2) showed that *Microdochium phyllosaprophyticum* sp. nov. formed an independent clade closely related to *M. bambusae* and *M. indocalami*. *M. phyllosaprophyticum* is distinguished from *M. bambusae* (SAUCC 1862-1) by 6/535, 15/836, 81/857 and 15/714 characters and from *M. indocalami* (SAUCC 1016) by 8/535, 14/836, 63/840 and 17/714 characters in ITS, LSU, RPB2 and TUB2 sequences, respectively. Morphologically, *M. phyllosaprophyticum* differs from *M. bambusae* and *M. indocalami* in conidia (7.5–15.0 × 2.0–3.5 μm vs. 13.0–17.0 × 2.5–3.5 μm vs. 13.0–15.5 × 3.5–5.5 μm) and in colony texture (light brown to creamy white with regular margins vs. gray white with regular margins vs. white with regular margins on PDA) [21]. For details, see Table 3.

**Figure 4 jof-09-01176-f004:**
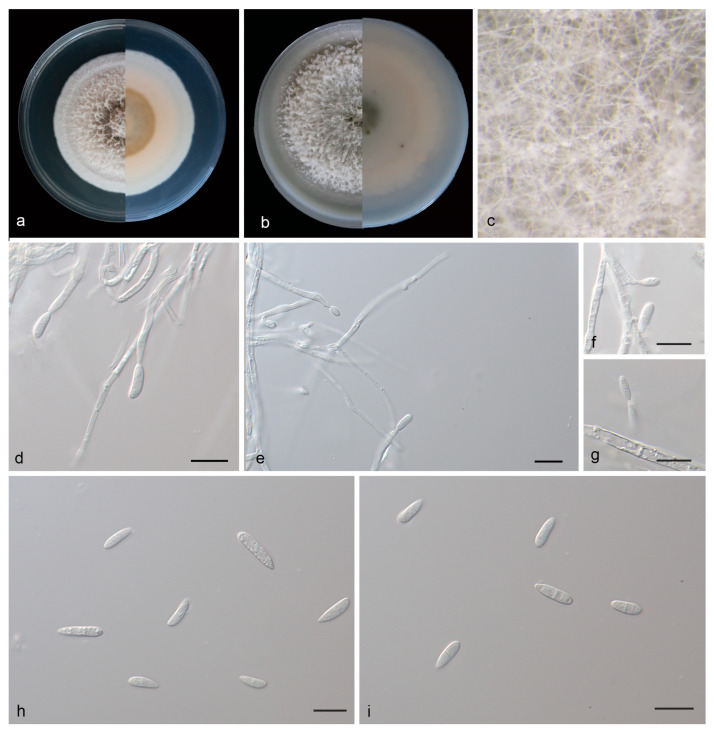
*Microdochium phyllosaprophyticum* (holotype, HMAS 352653). (**a**) Colonies on PDA from above and below after 14 days; (**b**) colonies on OA from above and below after 14 days; (**c**) colony overview; (**d**–**g**) conidiogenous cells and conidia; (**h**,**i**) conidia. Scale bars: (**e**–**i**) 10 μm.

**Table 3 jof-09-01176-t003:** The asexual morphological characters of some *Microdochium* species.

Species	Conidiogenous Cells	Size of Conidiogenous Cells (μm)	Conidia	Size of Conidia (μm)	References
*M* *icrodochium albescens*	Doliiform to obpyriform	6–15 × 1.5–4	Falcate, apex pointed	11–16 × 3.5–4.5	[13]
*M.* *bambusae*	Cylindrical and smooth	17.4–30.0 × 2.5–3.0	Oblong to ellipsoid	13.0–17.0 × 2.5–3.5	This study
*M. bolleyi*	Ampullate or cylindrical	3.1–6.4 × 2.5–3.8	Crescent, lunate	5.0–8.7 × 1.6–2.3	[13]
*M. chrysanthemoides*	Cylindrical to ellipsoidal	5–12 × 3.0–4.5	Ellipsoid or allantoid	4.5–7 × 2–3	[13]
*M. chuxiongense*	Solitary, clavate	27–74 × 2–3	Shuttle or sickle	4–12 × 2–5	[39]
*M. citrinidiscum*	Denticulate, cylindrical	11–29 × 1.5–2	clavate to obovoid	7–31 × 2–3	[13]
*M. colombiense*	Ampulliform, polyblastic	5–11.5 × 2.5–3.5	Fusiform, allantoid	5–8 × 1.5–2.5	[13]
*M. dawsoniorum*	Cylindrical to irregular	20–30 × 1–2	Flexuous to falcate	25–75 × 1–2	[13]
*M. fisheri*	Denticulate, cylindrical	19–60 × 1.5–2	Obovoid, subpyriform	7–12 × 3–4	[13]
*M. hainanense*	Ampulliform and lageniform	4.8–8.2 × 2.0–2.5	Spindle to rod shaped	7.0–16.1 × 2.5–4.7	[21]
*M. indocalami*	Cylindrical, straight or bent	11.0–28.3 × 1.5–2.9	Clavate to obovoid	13.0–15.5 × 3.5–5.5	[17]
*M. lycopodinum*	Ampulliform to lageniform	4–12 × 2.5–3.5	Fusiform or with one side	8–15 × 2.5–3.5	[13]
*M. maculosum*	Denticulate, raduliform	7–39 × 1–3	Fusiform, straight or curved	6–15 × 2–4	[18]
*M. miscanthi*	Smooth and cylindrical	9.7–14.5 × 3.6–4.1	Spindle to rod shaped	7.0–16.1 × 2.5–4.7	[21]
*M. musae*					
*M. nannuoshanense*	Cylindrical and smooth	19.0–27.0 × 2.0–3.0	Oblong to ellipsoid	7.0–12.0 × 2.5–4.0	This study
*M. neoqueenslandicum*	Lageniform to subcylindrical	4.5–10 × 2–3.5	Lunate, allantoid	4–9 × 1.5–3	[13]
*M. nivale*	Doliiform to obpyriform	6–15 × 2.2–4	Falcate, apex pointed	5–36 × 2–4.5	[13]
*M. paspali*	Lageniform to cylindrical	6.5–15.5 × 2.5–4	Falcate, apex pointed	7–20.5 × 2.5–4.5	[13]
*M. phyllosaprophyticum*	Cylindrical and smooth	16.0–32.0 × 2.0–2.5	Oblong to ellipsoid	7.5–15.0 × 2.0–3.5	This study
*M. phragmitis*	Ampulliform to lageniform	5–12(–30) × 2.5–3	Ellipsoid-fusiform	10–16 × 2–3.5	[13]
*M. poae*	Cylindrical or subcylindrical	1.5–6.5 ×1–2	Fusiform, ovoid, pyriform	3.5–8.5 ×2–3	[13]
*M. ratticaudae*	–	–	Fusoid, falcat	7-11 × 1.5-2.5	[13]
*M. rhopalostylidis*	Ampulliform	4–10 × 3–3.5	Fusoid, curved	16–20(–23) × 2.5–3	[15]
*M. seminicola*	Ampulliform to lageniform	7–9.5 × 3–4	Cylindrical to fusiform	19–54 × 3–4.5	[13]
*M. sinense*	Smooth and cylindrical	16.3–22.4 × 4.1–5.7	Spindle shaped or cylindrical	11.5–19.34 × 2.8–5.4	[21]
*M. sorghi*	Ampulliform to obclavate	5–13 × 3–4	Filiform, obclavate	20–90 × 1.5–4.5	[13]
*M. tainanense*	Cylindrical or ampulliform	3–10 × 1–3	Lunate	10–15 × 2–3	[13]
*M. trichocladiopsis*	Cylindrical to clavate	4–37 × 2–3	Oblong, fusiform to obovoid	6–18 × 2–3.5	[13]
*M. yunnanense*	Ampulliform, lageniform	6.5–10.0 × 2.5–3.4	Ellipsoid and cylindrical	6.8–10.0 × 2.4–3.5	[17]

## 4. Discussion

*Microdochium* was established by Syd. & P. Syd. in 1924; it has belonged to *Microdochiaceae* since 2016 (*Microdochiacea* was introduced by Hernández-Restrepo in 2016) [1,9,10,11,12]. *Monographella* Petr. was considered a sexual morph of the genus; “One Fungus = One Name” was proposed in The Amsterdam Declaration on Fungal Nomenclature in 2011, but the genus name of *Microdochium* was retained as the correct genus name [40]. With the development of molecular techniques, *Microdochium* was divided into a new family called *Microdochiaceae* by Hernández-Restrepo et al. [8]. *Microdochiaceae* is characterized by asexual morphs of polyblastic, sympodial or annellidic conidiogenous cells with hyaline conidia; the conidia come in a variety of shapes, i.e., cylindrical, fusiform, oval, rod-shaped, vertical or curved, truncated at the base and mostly rounded at the apex, and *Monographella*-like sexual morphs [8,21].

In the present study, six strains from one host plant (*Bambusaceae* sp.) and saprophytic leaves were split into three new species (*Microdochium bambusae*, *M. nannuoshanense* and *M. phyllosaprophyticum*). Currently, the Global Biodiversity Information Facility (GBIF, https://www.gbif.org/, accessed on 30 October 2023) contains 1693 georeferenced records of *Microdochium* species reported around the world. America, Asia, Europe and Oceania were the main distribution locations for this species; the United States has an especially great distribution, followed by Poland [2,8,13,14,15,19,20,41]. Since the 21st century, 12 species of this genus have been reported from five provinces (Fujian, Guizhou, Hainan, Sichuan and Yunnan) and Beijing in China (including the present study), viz, *Microdochium bambusae* sp. nov. (*Bambusaceae* sp.), *M. chrysanthemoides* (air), *M. chuxiongense* (*Bondarzewia* sp.), *M. graminearum* (decaying herbaceous grass stem), *M. hainanense* (*Phragmites australis*), *M. indocalami* (*Indocalamus longiauritus*), *M. miscanthi* (*Miscanthus sinensis*), *M. nannuoshanense* sp. nov. (*Bambusaceae* sp.), *M. paspali* (*Paspalum vaginatum*), *M. phyllosaprophyticum* sp. nov. (saprophytic leaves), *M. poae* (*Poa annua*), *M. shilinense* (decaying herbaceous stem of grass), *M. sinense* (*Miscanthus sinensis*) and *M. yunnanense* (*Indocalamus longiauritus*) [2,13,17,20,21,39,42]. Previous studies have shown that *Microdochium* species had been introduced on a range of host families, such as *Arecaceae*, *Asteraceae*, *Cactaceae*, *Euphorbiaceae*, *Iridaceae*, *Lycopodiaceae*, *Musaceae*, *Myrtaceae*, *Passifloraceae* and *Phyllanthaceae*, and more than half of *Microdochium* fungi were associated with *Poaceae* plants [42]. In particular, the study of *Bambusaceae*, as a subfamily of *Poaceae* plants, has become increasingly popular in recently years; an especially large number of *Apiospora* fungi have been isolated from *Bambusaceae* [43], whereas only a few *Microdochium* fungi have been isolated from *Bambusaceae*.

*Microdochium* species were reported as endophytes, plant pathogens and saprophytes. *M. bolleyi* was reported as a root endophyte and was proved to act against *Gaeumannomyces graminis* var. *tritici*, and it may be further used as a biocontrol agent [44]. Most species of *Microdochium* were reported as plant pathogens, especially against economical cereal crops: for example, *Microdochium albescens*, *M. oryzae* isolated from *Oryza sativa*; *M. majus*, *M. nivale*, *M. trichocladiopsis* and *M. triticicola* isolated from *Triticum aestivum*; and so on [42]. In reality, some species have not been regarded as plant pathogens; they are only isolated from leaf spot. However, the pathogenicity of this fungi has not been proven by Koch’s Postulates. Generally, we believe that the type of fungus is an endophytic fungus associated with leaf spot. In addition, we should focus more on disease detection and biological control.

## 5. Conclusions

In this study, we isolated six strains associated with *Microdochium* on *Bambusaceae* sp. and saprophytic leaves by tissue isolation and single spore isolation from Hainan and Yunnan provinces in China. Based on morphology and phylogeny, six strains were identified as three new species, viz., *Microdochium bambusae* sp. nov., *M. nannuoshanense* sp. nov. and *M. phyllosaprophyticum* sp. nov. In the future, we firmly believe that *Microdochium* species will be isolated from more plants around the world.

## Figures and Tables

**Table 1 jof-09-01176-t001:** The PCR primers, sequence and cycles used in this study.

Loci	PCR Primers	Sequence (5′–3′)	PCR Cycles	References
LSU	LR0R	GTA CCC GCT GAA CTT AAG C	(95 °C: 30 s, 51 °C: 60 s, 72 °C: 1 min) × 35 cycles	[27]
	LR5	TCC TGA GGG AAA CTT CG		
ITS	ITS5	GGA AGT AAA AGT CGT AAC AAG G	(95 °C: 30 s, 55 °C: 30 s, 72 °C: 30 s) × 35 cycles	[28]
	ITS4	TCC TCC GCT TAT TGA TAT GC		
RPB2	RPB2-5F	GAY GAY MGW GAT CAY TTY GG	(95 °C: 30 s, 56 °C: 30 s, 72 °C: 1 min) × 35 cycles	[29,30]
	RPB2-7CR	CCC ATW GCY TGC TTM CCC AT		
TUB2	Btub526_F	CGA GCG YAT GAG YGT YTA CTT	(95 °C: 30 s, 56 °C: 30 s, 72 °C: 45 s) × 35 cycles	[7]
	Btub1332_R	TCA TGT TCT TGG GGT CGA A		

**Table 2 jof-09-01176-t002:** GenBank accession numbers of the taxa used in phylogenetic reconstruction.

Species	Strain No.	Region	GenBank Accession No.			
			ITS	LSU	RPB2	TUB2
*Idriela lunata*	CBS 204.56 ^T^	USA	KP859044	KP858981	–	–
	CBS 177.57	USA	KP859043	KP858980	–	–
*Microdochium albescens*	CBS 243.83	Ivory Coast	KP858994	KP858930	KP859103	KP859057
	CBS 291.79	Ivory Coast	KP858996	KP858932	KP859105	KP859059
** *M.* *bambusae* **	**SAUCC 1862-1** ** ^T^ **	**China**	**OR702567**	**OR702576**	**OR715791**	**OR715785**
	**SAUCC 1866-1**	**China**	**OR702568**	**OR702577**	**OR715792**	**OR715786**
*M. bolleyi*	CBS 540.92	Syria	KP859010	KP858946	KP859119	KP859073
*M. chrysanthemoides*	CGMCC 3.17929 ^T^	China	KU746690	KU746736	–	–
*M. chuxiongense*	YFCC 8794 ^T^	China	OK586161	OK586160	OK584019	OK556901
*M. citrinidiscum*	CBS 109067 ^T^	Peru	KP859003	KP858939	KP859112	KP859066
*M. colombiense*	CBS 624.94	Colombia	KP858999	KP858935	KP859108	KP859062
*M. dawsoniorum*	BRIP 65649 ^T^	Australia	MK966337	–	–	–
*M. fisheri*	CBS 242.90 ^T^	UK	KP859015	KP858951	KP859124	KP859078
*M. graminearum*	CGMCC 3.23525 ^T^	China	OP103966	OP104016	OP236027	–
	CGMCC 3.23524	China	OP103965	OP104015	OP236026	–
*M. hainanense*	SAUCC 210782	China	OM956296	OM959324	OM981154	OM981147
	SAUCC 210781 ^T^	China	OM956295	OM959323	OM981153	OM981146
*M. indocalami*	SAUCC 1016 ^T^	China	MT199884	MT199878	MT510550	MT435653
*M. lycopodinum*	CBS 146.68	The Netherlands	KP858993	KP858929	KP859102	KP859056
	CBS 122885 ^T^	Germany	KP859016	KP858952	KP859125	KP859080
*M. maculosum*	COAD 3358 ^T^	Brazil	Ok966954	Ok966953	–	–
*M. majus*	CBS 741.79	Germany	KP859001	KP858937	KP859110	KP859064
*M. miscanthi*	SAUCC 211092 ^T^	China	OM956214	OM957532	OM981148	OM981141
	SAUCC 211093	China	OM956215	OM957533	OM981149	OM981142
*M. musae*	CPC 32809	Malaysia	MH107894	MH107941	–	–
	CBS 143500 ^T^	Malaysia	MH107895	MH107942	MH108003	–
** *M. nannuoshanense* **	**SAUCC 2450-1 ^T^**	**China**	**OR702569**	**OR702578**	**OR715793**	**OR715787**
	**SAUCC 2450-3**	**China**	**OR702570**	**OR702579**	**OR715794**	**OR715788**
*M. neoqueenslandicum*	CBS 445.95	The Netherlands	KP858997	KP858933	KP859106	KP859060
	CBS 108926 ^T^	The Netherlands	KP859002	KP858938	KP859111	KP859065
*M. nivale*	CBS 116205 ^T^	UK	KP859008	KP858944	KP859117	KP859071
*M. nivale var. majus*	CBS 177.29	The Netherlands	MH855031	MH866500	–	–
*M. nivale var. nivales*	CBS 288.50	The Netherlands	–	MH868135	–	–
*M. novae-zelandiae*	CPC 29376 ^T^	The Netherlands	LT990655	–	LT990641	LT990608
	CPC 29693	The Netherlands	LT990656	n/a	LT990642	LT990609
*M. paspali*	HK-ML-1371	China	KJ569509	–	–	KJ569514
	CBS 138620 ^T^	China	KJ569513	–	–	KJ569518
** *M. phyllosaprophyticum* **	**SAUCC 3583-1 ^T^**	**China**	**OR702571**	**OR702580**	**OR715795**	**OR715789**
	**SAUCC 3583-6**	**China**	**OR702572**	**OR702581**	**OR715796**	**OR715790**
*M. phragmitis*	CBS 285.71 ^T^	Poland	KP859013	KP858949	KP859122	KP859077
	CBS 423.78	Poland	KP859012	KP858948	KP859121	KP859076
*M. poae*	CGMCC3.19170 ^T^	China	MH740898	–	MH740906	MH740914
	LC 12115	China	MH740901	–	MH740909	MH740917
	LC 12116	China	MH740902	–	MH740910	MH740918
*M. ratticaudae*	BRIP 68298 ^T^	Australia	MW481661	MW481666	MW626890	–
*M. rhopalostylidis*	CBS 145125 ^T^	The Netherlands	MK442592	MK442532	MK442667	–
*M. seminicola*	CBS 139951 ^T^	Switzerland	KP859038	KP858974	KP859147	KP859101
	CPC 26001	Canada	KP859025	KP858961	KP859134	KP859088
	DAOM 250161	Canada	KP859034	KP858970	KP859143	KP859097
*M. shilinense*	CGMCC 3.23531 ^T^	China	OP103972	OP104022	–	OP242834
*M. sinense*	SAUCC 211097 ^T^	China	OM956289	OM959225	OM981151	OM981144
	SAUCC 211098	China	OM956290	OM959226	OM981152	OM981145
*M. sorghi*	CBS 691.96	Cuba	KP859000	KP858936	KP859109	KP859063
*M. tainanense*	CBS 269.76 ^T^	The Netherlands	KP859009	KP858945	KP859118	KP859072
	CBS 270.76	The Netherlands	KP858995	KP858931	KP859104	KP859058
*M. trichocladiopsis*	CBS 623.77 ^T^	The Netherlands	KP858998	KP858934	KP859107	KP859061
*M. yunnanense*	SAUCC 1011 ^T^	China	MT199881	MT199875	MT510547	MT435650
	SAUCC 1012	China	MT199882	MT199876	MT510548	MT435651

Notes: Ex-type or ex-epitype strains are marked with “^T^”, and the new species information described in this study is marked in bold. CBS: Westerdijk Fungal Biodiversity Institute; CGMCC: China General Microbiological Culture Collection; BRIP: Australian plant pathogen culture collection; LC: working collection of Dr. Lei Cai, housed at Institute of Microbiology, CAS, China; CPC: working collection of Pedro Crous maintained at the Westerdijk Institute.

## Data Availability

The sequences from the present study were submitted to the NCBI database (https://www.ncbi.nlm.nih.gov/, accessed on 20 October 2023), and the accession numbers are listed in Table 2.

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
