# Peer review of "Three New Species of *Microdochium* (*Microdochiaceae*, *Xylariales*) on *Bambusaceae* sp. and Saprophytic Leaves from Hainan and Yunnan, China"

_jof, 2023, doi:10.3390/jof9121176_

Round 1

Reviewer 1 Report

Comments and Suggestions for Authors

The manuscript:"Three new species of Microdochium (Microdochiaceae, 3 Xylariales) on Bambusaceae sp. and saprophytic leaves from 4 Hainan and Yunnan, China" deals with the problem of fungal species and proposes 3 new species of the genus Microdochium. In general, it is a scientifically very interesting manuscript, although austere. I will refer exclusively to the taxonomic problem and the verification of its premises.

1-The fundamental problem that I identify in the manuscript is the lack of a conceptual framework to approach the fungal species. It is evident that the authors use a phylogenetic premise through MLST and micromorphological characters to define the 3 new species. However, at least other phenotypic characteristics of the proposed species are needed to make this analysis more integrative. Phylogenetic resolution and morphological characteristics are insufficient without a precise conceptual framework. In my opinion, for this proposal to be incontestable, it needs an integrative approach with phenotypic and evolutionary-molecular responses (speciation test like coalescent models, distance partition, etc.). See minimal standards in: Aime et al. How to publish a new fungal species, or name, version 3.0. IMA Fungus 12, 11 (2021). Pay attention to section:

Species concepts. Although it is recommended to employ a combination of several experimental methods, referred to as a polyphasic approach or integrative taxonomy, for delimiting species, authors should provide a statement of the guiding species concept used to delimit newly proposed species (Lücking et al. 2020). Because the best species concepts to apply can vary, authors of new species should be familiar with the concept(s) that have been tested and applied to their group. One recommended strategy is to consult taxonomic experts for a given group or consider their collaboration in the description of new species.

2-I am not convinced that the taxon M. nannuoshanense SAUCC 2450 is really a new species; In fact, phylogenetically it does not show a significant transition that separates it from M. sinense. I reconstructed the phylogeny with the data provided by the authors in Supplementary Material 1 under the GTR evolutionary model; I recognize that the topology is preserved with respect to the tree that the authors present in their manuscript. However, when evaluating the evolutionary-molecular hypothesis using the bPTP speciation model (https://github.com/zhangjiajie/PTP), I obtain evidence that confirms my phylogenetic suspicion:

Isolates 2450_1 and 2450_2 do not contain significant speciation signatures under the Bayesian Poisson tree model, but rather form the same species (Bayessian support is significant). Here I show the prediction:

Although the isolates 1862 and 3583 are detected as an individual species, the support is quite discrete (~0.5 Bayesian support); which indicates that validation is needed by other models of speciation and integrative evidence of biological transitions that separate it more precisely (Genomics is a good option).

In summary, the authors should offer more support for their proposal from different experimental methods, not only MLST and micromorphology (phenotypic responses and speciation models is a good start...) and accommodate their proposal under an appropriate conceptual framework (see the hypothesis of the phylophenetic species concept in microorganisms).

Comments on the Quality of English Language

/

Author Response

Dear Editors,

Thank you for your valuable suggestion. In response to these questions, I answer as follows:

I accepted the suggestions and carefully read this article “How to publish a new fungal species, or name, version 3.0.”. In addition, I also referred to articles published in recent years about the genus of Microdochium. The latest article released this year about the genus of Microdochium was published on Phytotaxa: Dissanayake et al. Microdochium sichuanense sp. nov. (Microdochiaceae, Xylariales), from a Poaceae host in Sichuan, China (https://doi.org/10.11646/phytotaxa.600.3.7). Dissanayake et al. introduced a new species based on morphological characterization and phylogenetic analyses. This article used three genes (LSU, ITS and rpb2), lack of TUB2 gene. I have reviewed previous article and found that there are basically four genes, including TUB2. In this study, we combined four gene (LSU, ITS, TUB2 and RPB2), we revealed three new species from molecular means. And in terms of morphology, these three new species all have gaps with their related species including macroscopic morphology (colony size, color, growth rate) and microscopic morphology (meristem cells, sporulation apparatus).

You mentioned that this species (Microdochium nannuoshanense SAUCC 2450) is not a new species and obtained a result using supplementary data. However, the phylogenetic tree (ML and Bayes tree see bottom) we constructed indeed proves that it is a new species (More than 20 base differences). Leaving aside phylogenetic analysis, there is also a significant difference in morphology between these two species (M. nannuoshanense SAUCC 2450 and M. sinense SAUCC210097). There is a significant difference in spore size:

M. nannuoshanense differs from M. sinense in conidia (7.0‒12.0 × 2.5‒4.0 μm vs. 11.5‒19.34 × 2.8‒5.4 μm). In addition, there are differences in their colony morphology, color, and growth rate. In fact, both of these species come from our laboratory, and the two samples were taken at different times and from different sampling locations.

Your valuable feedback has greatly benefited us. You proposed genomics is a good option, which is indeed a current research hotspot. At present, there are only three genome data (NCBI) in this Microdochium, which is far from enough. Our laboratory is also conducting systematic omics sequencing work. In Microdochium, we have previously reported some new species, including this time, we will conduct genomic sequencing. Not so, we will also collect species belonging to this group from all over the world, and conduct unified omics sequencing and analysis work (single copy orthologous genes). We hope to use omics methods to better reveal the phylogenetic relationships of this Microdochium.

Finally, we greatly appreciate your valuable advice and guidance. The classification methods we have encountered in our laboratory are only polygenic, morphological, and genomic. We will try our best to learn and master other experimental methods or techniques to better adapt to the trend of the times. Finally, I wish you all the best.

Best wishes,

Zhaoxue Zhang

Reviewer 2 Report

Comments and Suggestions for Authors

Three new species of Microdochium (Microdochiaceae, Xylariales) on Bambusaceae sp. and saprophytic leaves from Hainan and Yunnan, China

The authors propose the introduction of three new species within the genus Microdochium with a phylogenetic analysis that shows good statistical support, good illustrations and a table that denotes the morphological differences between the different species of the genus. However, I suggest making some corrections that will help improve The manuscript. I indicate them below:

Introduction:

-          In Line 31, what is the meaning of “J+ funnel-shaped”?

-          In line 41: the scientific name “M. paspali” should be written in extense because is the first mention in the manuscript. The same should be corrected in line 43 and in the entire manuscript.

-          In line 51: ITS is mentioned for the first time in the introduction so should be written as “internal transcribed spacer of ribosomal RNA gene (ITS)” and in line 56 only mention as “ITS”.

-          In line 54: the sentence is wrong and must be corrected since they did not obtain their respective morphological characteristics by separation and purification, using sequences of four molecular markers.

-          In line 55: “Partial” should be written without capital letter.

-          In line 58: the sentence after “(TUB2).”, must begin with a capital letter.

Materials and Methods:

-          The methodology should be written in past, so the text must be corrected.

-          In line 69 the culture media should be writes as follow: Potato Dextrose Agar (PDA), the same for OA.

-          In line 79 add an space between the number (2.0) and the unit of measurement (mL)

-          In line 78 delete the space between ° C.

-          In line 80 “Digimizer” should be written without a capital letter.

-          In line 94 “partial nuclear ribosomal large subunit” should be deleted because it was written in full form in the introduction, the same for the others phylogenetical markers.

-          In line 97 delete “including sequence” and “polymerase chain reaction (PCR) cycles”.

-          In line 99 delete “Cat No. 10154ES03” and add the city and country.

-          In line 102, for GelRed add the information of brand, city and country like performed for primers.

-          Again, the information for electrophoresis should be written in past form. Correct this in all material and methods.

-          In line 105: The authors do not report how they edited the sequences, in what program, this must be mentioned. Along the same lines, the consensus sequence is not obtained in MEGA.

-          In table 2. The meaning of all the acronyms of the collection numbers, for example: CBS, CPC, etc., must be added in the footer of the table.

Results: the new species are phylogenetically well supported.

-          The authors begin the results item directly with the phylogenetic analysis. I recommend introducing information related to the percentages of identity with the ITS sequence when comparing it with those of databases such as the NCBI, they will probably be low percentages that would support or cause suspicion of a possible new species.

-          In legend of figure 1, delete “combined”, it is repeated.

-          In figure 1: I recommend putting Bold branches indicating a BIPP/ML of 1/100.

-          In figure 1 I recommend to indicate with “T” in a superscript indicating the type strains used in the phylogenetic tree. The same for the new species proposed, It should be indicated on the tree which corresponds to the type species.

-          In line 192 and 271: after “sp. nov”, write "figure 2" and “figure 4” respectively, like in line 232.

Comments on the Quality of English Language

Author Response

Dear Editors,

Thank you for your valuable suggestion. In response to these questions, I answer as follows:

Introduction:

  1. In Line 31, what is the meaning of “J+ funnel-shaped”?

I deleted it.

  1. In line 41: the scientific name “M. paspali” should be written in extense because is the first mention in the manuscript. The same should be corrected in line 43 and in the entire manuscript.

I accepted the suggestions.

  1. In line 51: ITS is mentioned for the first time in the introduction so should be written as “internal transcribed spacer of ribosomal RNA gene (ITS)” and in line 56 only mention as “ITS”.

I accepted the suggestions.

  1. In line 54: the sentence is wrong and must be corrected since they did not obtain their respective morphological characteristics by separation and purification, using sequences of four molecular markers.

I accepted the suggestions.

  1. In line 55: “Partial” should be written without capital letter.

I accepted the suggestions.

  1. In line 58: the sentence after “(TUB2).”, must begin with a capital letter.

I accepted the suggestions.

Materials and Methods:

  1. The methodology should be written in past, so the text must be corrected.

I accepted the suggestions.

  1. In line 69 the culture media should be writes as follow: Potato Dextrose Agar (PDA), the same for OA.

I accepted the suggestions.

  1. In line 79 add an space between the number (2.0) and the unit of measurement (mL)

I accepted the suggestions.

  1. In line 78 delete the space between ° C.

I accepted the suggestions.

  1. In line 80 “Digimizer” should be written without a capital letter.

I accepted the suggestions and revised it.

  1. In line 94 “partial nuclear ribosomal large subunit” should be deleted because it was written in full form in the introduction, the same for the others phylogenetical markers.

I accepted the suggestions and revised it.

  1. In line 97 delete “including sequence” and “polymerase chain reaction (PCR) cycles”.

I accepted the suggestions and revised it.

  1. In line 99 delete “Cat No. 10154ES03” and add the city and country.

I accepted the suggestions and revised it, but, the “Cat No. 10154ES03” is required by the merchant to be written in this way, as it will reward us.

  1. In line 102, for GelRed add the information of brand, city and country like performed for primers.

I accepted the suggestions and revised it.

  1. Again, the information for electrophoresis should be written in past form. Correct this in all material and methods.

I accepted the suggestions and revised it.

  1. In line 105: The authors do not report how they edited the sequences, in what program, this must be mentioned. Along the same lines, the consensus sequence is not obtained in MEGA.

I accepted the suggestions and revised it.

  1. In table 2. The meaning of all the acronyms of the collection numbers, for example: CBS, CPC, etc., must be added in the footer of the table.

I accepted the suggestions and revised it.

Results: the new species are phylogenetically well supported.

  1. The authors begin the results item directly with the phylogenetic analysis. I recommend introducing information related to the percentages of identity with the ITS sequence when comparing it with those of databases such as the NCBI, they will probably be low percentages that would support or cause suspicion of a possible new species.

I accepted the suggestions.

  1. In legend of figure 1, delete “combined”, it is repeated.

I accepted the suggestions and revised it.

  1. In figure 1: I recommend putting Bold branches indicating a BIPP/ML of 1/100.

I accepted the suggestions and revised it.

  1. In figure 1 I recommend to indicate with “T” in a superscript indicating the type strains used in the phylogenetic tree. The same for the new species proposed, It should be indicated on the tree which corresponds to the type species.

I accepted the suggestions and revised it.

  1. In line 192 and 271: after “sp. nov”, write "figure 2" and “figure 4” respectively, like in line 232.

I accepted the suggestions and revised it.

Best wishes,

Zhaoxue Zhang

Reviewer 3 Report

Comments and Suggestions for Authors

General comments

The manuscript presents the isolation and characterization of three new species from the genus Microdochium, named M. bambusae, M. nannuoshanense and M. phyllosaprophyticum, derived from diseased leaves of Bambusaceae sp. and saprophytic leaves in Yunnan Province, China. The species identification is a result of a multi-locus phylogenetic analyses of 56 strains from the genus Microdochium based on combined dataset of ITS, LSU, RPB2 and TUB2 sequences. A detailed morphological characterization of the new species was also conducted.

The study is well-structured and the topic of investigation is relevant to the scope of Journal of Fungi. The new data leads to enhancement of the genus Microdochium with three novel members, each of them represented by two strains. All taxonomic features are described in detail. Morphological characteristics that distinguish the new representatives of the genus from closely related species are pointed by the authors of the paper.

Although comprehensive experiments have been carried out, there are unclear sentences in all parts of the text that need to be corrected in the revise version of the manuscript.

Specific comments

The Abstract is minimalistic, but the second sentence is too long, respectively not clear enough. It would be better to split it into two separate sentences.

The Introduction is informative and well-structured.

In general, the Materials and methods are presented in detail, but they are not clearly described in plain sentences.

Row 64 - unclear sentence

Rows 64-71 – This paragraph should be synchronized with the tense used in the rest of the text. The same for the sentence from rows 101-103.

Description of the section 2.3 Phylogenetic analyses is uncertain. It should be rewritten with more accurate phrases.

There are unclear sentences in the Results section too. For instance, the sentences on row 146, rows 161-162 from the paragraph 3.1. Phylogenetic analyses.

Row 215 – “Phylogenetic analyses of three combined genes (ITS, LSU, RPB2 and TUB2)…” - need of correction.

The Discussion section is in the context of phylogeny and taxonomy of the genus Microdochium, origin of new species and host plants. It is informative, but again there are unclear phrases that need to be revised (rows 316-320, 326-329, 345-349).

The last paragraph should be rewritten too. Some inaccuracies are listed below and are highlighted:

Rows 351-352 – “As endophytes, M. bolleyi was reported as root endophytes, and for the mycobiota of Iran was proved to against Gaeumannomyces graminis var. tritici” – need of correction

Rows 352-353 – “Most species of Microdochium were reported as plant pathogens, especially economical cereal crops…” – should be “….especially against economical cereal crops…”

Rows 356-358 – “…some species weren’t true “plant pathogens”, it is only isolated from the leaf spot…” - – need of correction

Rows 364-365 – “…six strains were identified three new species…”, will be better to be “…six strains were identified as three new species…”

The Conclusions are supported by the results.

All tables are informative and well constructed, however the figures need of some corrections.

Figure 1 – there is a repetition in the title: “A Maximum Likelihood tree based on combined a combined dataset of analyzed ITS, ….”

Figures 3a – the photo is too dark and disease symptoms on a leaf of Bambuaceae sp. are not visible, in contrast to Figure 2a, where the picture of the bamboo leaf is bigger and symptoms are evident.

The list of References contains mainly recent publications.

Comments on the Quality of English Language

English language required some improvements that are pointed above. 

Author Response

Dear Editors,

Thank you for your valuable suggestion. In response to these questions, I answer as follows:

  1. The Abstract is minimalistic, but the second sentence is too long, respectively not clear enough. It would be better to split it into two separate sentences.

I accepted the suggestions.

  1. The Introduction is informative and well-structured.

Thanks.

  1. In general, the Materials and methods are presented in detail, but they are not clearly described in plain sentences.

I accepted the suggestions.

  1. Row 64 - unclear sentence

I accepted the suggestions and revised it “A specimen usually isolates multiple fungi, and we obtain a single colony through single spore isolation and tissue isolation methods”.

  1. Rows 64-71 – This paragraph should be synchronized with the tense used in the rest of the text. The same for the sentence from rows 101-103.

I accepted the suggestions and revised it.

  1. Description of the section 2.3 Phylogenetic analyses is uncertain. It should be rewritten with more accurate phrases.

I accepted the suggestions and removed a portion.

  1. There are unclear sentences in the Results section too. For instance, the sentences on row 146, rows 161-162 from the paragraph 3.1. Phylogenetic analyses.

I accepted the suggestions and revised it.

  1. Row 215 – “Phylogenetic analyses of three combined genes (ITS, LSU, RPB2 and TUB2)…” - need of correction.

I accepted the suggestions and revised it.

  1. The Discussion section is in the context of phylogeny and taxonomy of the genus Microdochium, origin of new species and host plants. It is informative, but again there are unclear phrases that need to be revised (rows 316-320, 326-329, 345-349).

I accepted the suggestions and revised it.

The last paragraph should be rewritten too. Some inaccuracies are listed below and are highlighted:

  1. Rows 351-352 – “As endophytes, M. bolleyi was reported as root endophytes, and for the mycobiota of Iran was proved to against Gaeumannomyces graminis var. tritici” – need of correction

I accepted the suggestions and revised it.

  1. Rows 352-353 – “Most species of Microdochium were reported as plant pathogens, especially economical cereal crops…” – should be “….especially against economical cereal crops…”

I accepted the suggestions and revised it.

  1. Rows 356-358 – “…some species weren’t true “plant pathogens”, it is only isolated from the leaf spot…” - – need of correction

I accepted the suggestions and revised it.

  1. Rows 364-365 – “…six strains were identified three new species…”, will be better to be “…six strains were identified as three new species…”

I accepted the suggestions and revised it.

The Conclusions are supported by the results.

All tables are informative and well constructed, however the figures need of some corrections.

  1. Figure 1 – there is a repetition in the title: “A Maximum Likelihood tree based on combined a combined dataset of analyzed ITS, ….”

I accepted the suggestions and revised it.

  1. Figures 3a – the photo is too dark and disease symptoms on a leaf of Bambuaceae sp. are not visible, in contrast to Figure 2a, where the picture of the bamboo leaf is bigger and symptoms are evident.

I accepted the suggestions and revised it.

Best wishes,

Zhaoxue Zhang